# Knockdown of Mitogen-Activated Protein Kinase Kinase 3 Negatively Regulates Hepatitis A Virus Replication

**DOI:** 10.3390/ijms22147420

**Published:** 2021-07-10

**Authors:** Tatsuo Kanda, Reina Sasaki-Tanaka, Ryota Masuzaki, Naoki Matsumoto, Hiroaki Okamoto, Mitsuhiko Moriyama

**Affiliations:** 1Division of Gastroenterology and Hepatology, Department of Medicine, Nihon University School of Medicine, 30-1 Oyaguchi-kamicho, Itabashi-ku, Tokyo 173-8610, Japan; sasaki.reina@nihon-u.ac.jp (R.S.-T.); masuzaki.ryota@nihon-u.ac.jp (R.M.); matsumoto.naoki@nihon-u.ac.jp (N.M.); moriyama.mitsuhiko@nihon-u.ac.jp (M.M.); 2Division of Virology, Department of Infection and Immunity, Jichi Medical University School of Medicine, Shimotsuke, Tochigi 329-0498, Japan; hokamoto@jichi.ac.jp

**Keywords:** HAV, MAP2K3, Toll-like receptor, zinc chloride

## Abstract

Zinc chloride is known to be effective in combatting hepatitis A virus (HAV) infection, and zinc ions seem to be especially involved in Toll-like receptor (TLR) signaling pathways. In the present study, we examined this involvement in human hepatoma cell lines using a human TLR signaling target RT-PCR array. We also observed that zinc chloride inhibited mitogen-activated protein kinase kinase 3 (MAP2K3) expression, which could downregulate HAV replication in human hepatocytes. It is possible that zinc chloride may inhibit HAV replication in association with its inhibition of MAP2K3. In that regard, this study set out to determine whether MAP2K3 could be considered a modulating factor in the development of the HAV pathogen-associated molecular pattern (PAMP) and its triggering of interferon-β production. Because MAP2K3 seems to play a role in antiviral immunity against HAV infection, it is a promising target for drug development. The inhibition of MAP2K3 may also prevent HAV patients from developing a severe hepatitis A infection.

## 1. Introduction

The hepatitis A virus (HAV) is one of the major causes of acute hepatitis [1,2,3], sometimes resulting in acute liver failure, leading to a transplant or even death [3,4]. Although a vaccine for HAV infection has been achieved, gaining access to vaccinations may be difficult in low- and middle-income countries due to the associated high costs or ethical problems [5,6]. If hygiene and sanitation are continuously improved but low anti-HAV immunity is maintained, future hepatitis A outbreaks may occur in places like Japan [7,8]. Therefore, it is important to develop antiviral drugs and to use HAV vaccines more widely.

We and others have demonstrated that zinc compounds (zinc sulfate, zinc chloride, zinc oxide and zinc oxide nanoparticles) have anti-HAV properties [9,10,11]. For one thing, zinc sulfate inhibits HAV replication while enhancing the expression of glucose-regulated protein 78 (GRP78/Bip), an antiviral that impedes HAV [12,13] in a dose-dependent manner [9,10]. In addition, zinc chloride inhibits HAV replication with or without interferon [10].

Toll-like receptor (TLR) signaling pathways are involved in the production of interferon and inflammatory cytokines [14,15] as part of the antiviral immune response. It has been reported that free zinc ions differentially regulate the TLR-dependent myeloid differentiation primary response 88 (MYD88), as well as Toll/IL-1R domain-containing adapter-inducing IFN-β (TRIF) signaling pathways [15]. Tumor necrosis factor (TNF) receptor-associated factor 6 (TRAF6) and transforming growth factor β (TGF-β) activated kinase 1 (TAK1) play important roles as molecular bridges in innate immunity by linking the upstream TLRs with the downstream mitogen-activated protein kinase (MAPK) and nuclear factor kappa B (NF-κB) signaling pathways [16]. The silencing of TRAF6 and TAK1 downregulates the LPS-induced phosphorylation of p38 MAPK and the c-Jun N-terminal kinase (JNK) [16]. The N-terminal, which is a really interesting new gene (RING) domain, and the first zinc finger domain of TRAF6 are essential for signaling by cytokines which are dependent upon TRAF6 [17].

MAPK kinase 3 (MAP2K3), activated by inflammation and endoplasmic reticulum (ER) stress [18], phosphorylates and activates p38 MAPK. MAP2K3 can be activated by insulin, and is required for the expression of glucose transporters [19]. Little is known about the effects of zinc chloride on the TLR signaling pathways in hepatocytes. In this study, we observed that zinc chloride inhibits MAP2K3 expression, and that the silencing of MAP2K3 could negatively regulate HAV replication in human hepatocytes.

## 2. Results

### 2.1. Effects of Zinc Chloride on the TLR Signaling Pathways in Huh7 Cells

We previously reported that a treatment of 5 μM zinc chloride for 72 h suppressed HAV genotype IIIA HA11-1299 replication by 62.2% compared with the untreated control in human hepatoma Huh7 cells [10]. We also observed the additive effects of zinc chloride on the suppression of HAV replication by interferon-α-2a in Huh7 cells [10]. It is well known that TLR signaling pathways are involved in the eradication of RNA viruses [20]. Here, we performed TLR signaling pathway-related gene expression profiles in Huh7 cells using real-time RT–PCR-based focused microarrays. A comparison of the TLR signaling pathway-related genes between Huh7 cells treated with or without 5 μM zinc chloride for 24 h is shown in Table 1. In order to investigate innate immunity, including TLRs, we treated the cells for only 24 h. Out of the 84 TLR signaling pathway-related genes examined, 4 (4.8%) and 9 (10.7%) genes were significantly upregulated or significantly downregulated, respectively (*n* = 3 for each set; *p* < 0.05).

### 2.2. Effects of Zinc Chloride on TLR Signaling Pathways in PLC/PRF/5 Cells

Furthermore, we performed TLR signaling pathway-related gene expression profiles in human hepatoma PLC/PRF/5 cells, using real-time RT-PCR-based focused microarrays. A comparison of the TLR signaling pathway-related genes between PLC/PRF/5 cells treated with or without 5 μM zinc chloride for 24 h is shown in Table 2. Out of the 84 TLR signaling pathway-related genes examined, 11 (13.1%) and 3 (3.6%) genes were significantly upregulated and significantly downregulated, respectively (*n* = 3 for each set; *p* < 0.05). Interleukin 1B (IL1B) tended to be upregulated (4.12-fold; *p* = 0.050819).

When we compared zinc chloride-regulated genes between Huh7 and PLC/PRF/5 cells, no genes upregulated by zinc chloride were found in either cell line (Figure 1a). However, we observed that MAP2K3 was the only significantly downregulated gene in both cell lines (Figure 1b). Therefore, we focused next on the association between MAP2K3 and HAV replication.

### 2.3. Knockdown of MAP2K3 Negatively Regulates HAV Replication

#### 2.3.1. Knockdown of MAP2K3 by siRNA Did Not Reduce the Viability of Huh7 Cells

The induction factors of the genes studied following the treatment of human hepatoma cell lines with zinc chloride ranged from 1.07 to 1.45, which seemed relatively low, as did the downregulation factors: all were less than 2, even if they were statistically significant (Table 1 and Table 2). Compared to the control cells, MAP2K3 mRNA expression was reduced by 10 and 14% in Huh7 and PLC/PRF/5 cells, respectively, following treatment with 5 μM zinc chloride (Table 1 and Table 2). We also examined whether MAP2K3 protein expression was downregulated in Huh7 cells treated with 5 μM zinc chloride for 24 h (Figure 2a,b). Of note, zinc chloride downregulated MAP2K3 protein expression by 40.1% compared to the mock control. The discrepancy of the reduction after the zinc chloride treatment between the mRNA and protein levels may be associated with the ubiquitin–proteasome pathway [21].

Next, we transfected siRNA specifically targeting MAP2K3 (si-MAP2K3) into Huh7 cells. Cell lysates were prepared for a Western blot analysis to detect the endogenous expression of MAP2K3 using a specific antibody. We observed an inhibition of MAP2K3 expression in Huh7 cells transfected with si-MAP2K3 compared to the control siRNA (si-C) (Figure 2c,d). Huh7 cells transfected with si-MAP2K3 exhibited no significant reduction in cell viability after 72 h of transfection when compared with the control.

#### 2.3.2. Effects of the Knockdown of MAP2K3 by Specific Small Interfering RNA (siRNA) on the HAV Subgenomic Replicon Replication in HuhT7 Cells

We co-transfected HAV subgenomic replicon and si-MAP2K3 or si-C into HuhT7 cells, which were stably expressing T7 RNA polymerase [22]. After 72 h of transfection, the cell lysates were prepared for a reporter assay to examine the HAV subgenomic replicon replication (Figure 2e). We observed that the transfection of si-MAP2K3 suppressed the HAV subgenomic replicon genotype IB replication by 48.8% when compared with si-C-transfected HuhT7 cells (Figure 2e).

#### 2.3.3. Effects of the Knockdown of MAP2K3 by Specific siRNA on the HAV Genotype IIIA HA11-1299 Replication in Huh7 cells

We transfected si-MAP2K3 or si-C, and after 24 h of transfection we infected the HAV genotype IIIA HA11-1299 at a 0.01 multiplicity of infection (MOI) into Huh7 cells. After 48 h of infection, the cellular RNA was extracted for an RT-PCR test to examine the HAV RNA levels (Figure 2f). We observed that the transfection of si-MAP2K3 suppressed HAV genotype IIIA HA11-1299 RNA by 45.3% when compared with si-C-transfected Huh7 cells (Figure 2f). Thus, the knockdown of MAP2K3 could result in a decrease in HAV subgenomic and genomic RNA replication.

### 2.4. Inhibition of MAP2K3 Enhances the Interferon-β Promoter Activities Stimulated by Polyinosinic–Polycytidylic Acid (poly(I:C)) and Suppresses HAV Replication

#### 2.4.1. Knockdown of MAP2K3 Enhances the Interferon-β Promoter Activities Stimulated by Poly(I:C)

Next, we studied the possibility that MAP2K3 could be a modulation factor in cell stimulation by a synthetic double-stranded RNA poly(I:C), which is used experimentally to model viral infections in vivo and in vitro [20]. We examined the effects of a knockdown MAP2K3 on IFN-β promoter activity transfected with or without poly(I:C). As it is well known that Huh7 cells have non-functional TLR3, we transfected plasmid DNA encoding the firefly luciferase gene under the control of the IFN-β promoter (IFN-β-luc) and si-MAP2K3 or si-C, with or without poly(I:C). At 16 h post-transfection, cells were collected and the IFN-β promoter activity was measured by reporter assay (Figure 3). Poly(I:C) seemed to have stimulated the IFN-β promoter activity mainly through the cellular retinoic acid-inducible gene I (RIG-I) and melanoma differentiation-associated protein 5 (MDA5) in Huh7 cells [20]. Notably, we observed that the knockdown of MAP2K3 could enhance the interferon-β promoter activity stimulated by poly(I:C) in both immortalized human hepatocytes (IHH) (Figure 3a) and Huh7 cells (Figure 3b). Thus, it is possible that MAP2K3 could be one of the modulation factors by which HAV pathogen-associated molecular pattern (PAMP) triggers the production of interferon-β.

#### 2.4.2. SB202190, a Potent Inhibitor of the MAP2K3–p38 MAPK Signaling Pathway, Enhances the Interferon-β Promoter Activities Stimulated by Poly(I:C)

Next, we examined the effects of SB202190, a potent and selective p38 MAP kinase inhibitor, on the IFN-β promoter activity transfected with or without poly(I:C). We transfected the plasmid IFN-β-luc with poly(I:C) into IHH cells, and after 24 h of transfection we added 10 nM SB202190. After 40 h of transfection, cells were collected and the IFN-β promoter activity was measured by reporter assay (Figure 4). We observed that SB202190 enhanced interferon-β promoter activity which was stimulated by poly(I:C) in IHH cells (Figure 4a) and tended to enhance the interferon-β promoter activity stimulated by poly(I:C) in Huh7 cells (Figure 4b). Thus, a chemical inhibitor against the MAP2K3–p38 MAPK signaling pathway also enhanced the interferon-β promoter activity stimulated by poly(I:C) in human immortalized human hepatocytes. We also observed that 5 nM SB202190 inhibited HAV genotype IIIA HA 11-1299 replication in both IHH and Huh7 cells (Figure 5a,b).

## 3. Discussion

In the present study, we revealed the following: (i) zinc chloride affects the TLR signaling pathways in human hepatocytes; (ii) MAP2K3 expression is downregulated by zinc chloride; and (iii) HAV replication is downregulated by the silencing MAP2K3 expression. Our results explained one of the different mechanisms of the inhibitory effects of zinc compounds on HAV replication from previous reports [9,10].

A higher risk of SARS-CoV-2 infection and its severe conditions coincide with patients with chronic diseases and the elderly, both of whom are at risk of zinc deficiency [23]. Zinc supplementation may potentially reduce the risk, duration and severity of infectious diseases, such as SARS-CoV-2 [24]. Zinc ions seemed to be profoundly involved in TLR signaling pathways, and they differentially regulate MYD88 and TRIF signaling via a zinc signal or via basal zinc ion levels, respectively [15]. Zinc-finger antiviral protein (ZAP) acted on RNA granules as a cytosolic RNA sensor via the exosome, which is an RNA degradation system [24]. ZAP is also known as a regulator of RIG-I signaling in human cell lines. RIG-I-like receptors (RLRs), RIG-I and MDA5 detect various RNA viruses, including HAV, in the cytosol and induce a type I IFN-dependent antiviral response [25]. 

HAV 3ABC, which is located specifically in the mitochondria, directs the proteolysis of the mitochondrial antiviral signaling protein (MAVS) [25]. The HAV 2B protein suppresses interferon-β gene transcription by interferon regulatory factor 3 (IRF3) activation [26]. The TRIF is proteolytically cleaved by HAV 3CD [27]. It is unknown whether HAV proteins induce the activation of NF-κB [28,29]. In human hepatocytes, HAV interacts with TLR signaling pathways, which are involved in the development of HAV pathogenesis [30]. HAV 2C induces a rearrangement of intracellular membranes, which are presumed to be ER-derived and the site of RNA synthesis. 

Of interest was the ER chaperon GRP78, which worked as an antiviral for HAV [12,13,31]. ER stress is an important factor in hepatic cell damage, which is induced by an innate immune response, including TLR signaling pathways [32]. Zinc sulfate inhibited HAV replication and enhanced GRP78 expression in a dose-dependent manner, although metallothioneins in a conditioned medium were enhanced by zinc sulfate in a dose-independent manner [9]. Zinc chloride enhanced the GRP78 levels in a conditioned medium from interferon-treated Huh7 cells infected with HAV (9.873 ng/mL compared to those mono-treated with interferon (5.531 ng/mL) or those which were untreated (1.544 ng/mL)) ([10] and unpublished data). Metallothioneins are small, cysteine-rich proteins capable of binding divalent cations, such as zinc and copper, and have long been classified as interferon-stimulated genes (ISGs), although metallothioneins are highly induced by zinc [33]. 

Late in the human cytomegalovirus infection period, MAP2K3 is increased and p38 MAPK is activated [34]. MicroRNA-21 pretreatment remarkably inactivated MAP2K3–p38 MAPK signaling and protected against coxsackievirus B3 infection in mice [35]. We also demonstrated that siRNA against MAP2K3 inhibited HAV replication (Figure 2). Chen et al. also reported that the activation of MAPK and Fos proto-oncogene (FOS) resulted in the upregulation of miR-21, which downregulated MyD88 and interleukin 1 receptor associated kinase 1 (IRAK1), and led to the inhibition of type I IFN production in Huh7 cells [36]. Our results support these findings [36]. Therefore, MAP2K3 seems to play a role in the antiviral immunity toward HAV infection.

In Japan, zinc chloride is not in clinical use for acute HAV infection. The normal serum zinc concentration is from 122 to 199 μM [37]. In Japanese daily clinical practice, patients with a zinc deficiency or Wilson’s disease take zinc acetate, including 50–100 or 150 mg daily zinc. Zinc plays a role in the activation and structural maintenance of as many as 300 proteins and enzymes which contribute to various biological processes, and zinc deficiency is often observed in patients with liver diseases [37]. Acute HAV infection occasionally leads to acute liver failure with coagulopathy and hyperammonemia [38]. Zinc supplementation is also useful for improved hyperammonemia [37].

We also demonstrated that si-MAP2K3 and SB202190 enhanced the interferon-β promoter activities stimulated by poly(I:C) (Figure 3 and Figure 4). MAP2K3 could be one of the modulation factors for the HAV PAMP triggering of interferon-β production. A previous study [20] showed an interferon bioassay using IHH cells. Unfortunately, we did not perform an interferon ELISA or bioassay, which would have reflected more precise bioactive interferon (Figure 3). 

As Huh7 cells are not known to produce significant amounts of type I interferons, the inhibition in HAV replication seen in these cells when treated with SB202190 may also involve some other mechanism (Figure 5b). These results in Huh7 cells were reinforced by the insignificant increase in the interferon-β reporter activity in Huh7 cells when they were treated with SB202190 (Figure 4b). Johnson et al. reported that SB202190 impaired the viral entry and reduced the cytokine induction by Zair ebolavirus in human dendritic cells [39]. The inhibition of JNK1/2 and p38 kinases by SB202190 resulted in the significant reduction of porcine circovirus type 2 (PCV2) viral mRNA transcription and protein synthesis, and the viral progeny release and blockage of the apoptosis of PCV2-infected cells [40]. Further studies will be needed in this regard.

## 4. Materials and Methods

### 4.1. Cell Culture and Reagents

The human hepatoma cell lines Huh7 and PLC/PRF/5 were maintained in Dulbecco’s modified Eagle’s medium (DMEM; Sigma-Aldrich, Saint Louis, MO, USA) containing 10% fetal calf serum (FCS), 100 U/mL penicillin G and 200 μg/mL streptomycin at 37 °C in an atmosphere of 5% CO_2_ [9]. As Huh7 and PLC/PRF/5 cells support HAV replication [9,33,41], we selected these cell lines for this study. HuhT7 cells, which are stably transformed derivates from Huh7, expressed the T7 RNA polymerase, and were maintained in DMEM (Sigma-Aldrich) containing 10% FCS, 100 U/mL penicillin G and 200 μg/mL streptomycin at 37 °C in an atmosphere of 5% CO_2_ [22]. The HuhT7 cells were kindly gifted from Prof. Gauss-Müller, Ph.D., University of Lübeck, Germany. The zinc chloride was purchased from Wako Pure Chemical (Tokyo, Japan). As cancer cells and cell lines may be more sensitive to the silencing of MAP2K3 [42], we also performed several experiments in human immortalized human hepatocytes (IHHs) to rule out the possibility that the phenotype is not a cell line artefact. IHHs, which were adopted in DMEM, were used [20]. Small interfering RNA (siRNA) against MAP2K3 (si-MAP2K3/si-MEK3) [sc-35907] and the control siRNA (si-C) [sc-37007] were purchased from Santa Cruz Biotechnology (Santa Cruz, CA, USA). SB202190, a potent and selective p38 MAPK inhibitor, was purchased from Selleck Chemicals (Houston, TX, USA).

### 4.2. HAV Subgenomic Replicon, its Transfection and Reporter Assay

Plasmid pT7-18f-LUC, a replication-competent HAV replicon containing an open reading frame with firefly luciferase flanked by the first four amino acids of the HAV polyprotein and by 12 C-terminal amino acids of VP1, followed by the P2 and P3 domains of the HAV polyprotein (HAV genotype IB strain HM175 18f, GenBank Accession No. M59808) were also kindly provided by Prof. Gauss-Müller, Ph.D., University of Lübeck, Germany. Approximately 1.0 × 10^5^ HuhT7 cells were seeded on 6-well tissue culture plates (Iwaki Glass, Tokyo, Japan) 24 h prior to the transfection. Plasmid pT7-18f-LUC (0.2 μg/well) with 50 μM si-MAP2K3 or 50 μM si-C were transfected into HuhT7 using Effectene transfection reagent (Qiagen, Hilden, Germany) according to the manufacturer’s instructions [13]. After 72 h of transfection, the cells were harvested using a reporter lysis buffer (Promega, Madison, WI, USA), and the firefly luciferase activities were determined by a Picagene system (Toyo Ink, Tokyo, Japan) using a luminometer (Luminescencer-JNR II AB-2300, ATTO, Tokyo, Japan).

### 4.3. HAV Infection

Approximately 1.0 × 10^5^ Huh7 cells were seeded on 6-well tissue culture plates (Iwaki Glass, Tokyo, Japan) 24 h prior to transfection. A total of 50 μM si-MAP2K3 or 50 μM si-C were transfected into Huh7 cells using Effectene transfection reagent (Qiagen), according to the manufacturer’s instructions [13]. After 24 h of transfection, the cells were infected with 0.01 MOI of HAV genotype IIIA HA11-1299 after the cells were washed with phosphate-buffered saline (PBS) twice. After 24 h of infection, the cells were washed with PBS and maintained in 5% FCS DMEM, with or without zinc chloride (Wako). After 48 h of infection, the cellular RNA was extracted in order to determine the HAV RNA [10].

### 4.4. Cell Viability Assay

In order to evaluate the cell growth and cell viability 72 h after the transfection of the siRNAs, a dimethylthiazol carboxymethoxyphenyl sulfophenyl tetrazolium (MTS) assay was performed using the CellTiter 96 Aqueous One-Solution Cell Proliferation Assay Kit (Promega) according to the manufacturer’s instructions. The absorbance at 490 nm of each well was measured with the iMark Microplate Absorbance Reader (Bio-Rad, Hercules, CA, USA).

### 4.5. Western Blotting

Cells were collected in a sodium dodecyl sulfate (SDS) lysis buffer. After sonication, the proteins were subjected to electrophoresis in a 5–20% SDS–polyacrylamide gel, and then transferred onto a polyvinylidene difluoride membrane (ATTO), followed by overnight blocking with 5% skimmed milk in phosphate-buffered saline with Tween 20 (Bio-Rad, Hercules, CA, USA). The membrane was probed with antibodies specific to MAP2K3 (Cell Signaling Technology, Boston, MA, USA), GAPDH (Santa Cruz), and/or *β*-tubulin (Abcam K.K., Tokyo, Japan). After washing the membrane, it was incubated with secondary horseradish peroxidase-conjugated antibodies for an hour. The signals were detected with enhanced chemiluminescence (GE Healthcare, Tokyo, Japan) and scanned with an image analyzer LAS-4000 mini (Fuji Film, Tokyo, Japan), as previously described [13].

### 4.6. RNA Extraction, cDNA Synthesis and Human TLR Signaling Targets PCR Array

Approximately 1.0 × 10^5^ Huh7 or PLC/PRF/5 cells per well were plated into a 6-well plate and, 24 h later, were treated with or without 5 μM zinc chloride (Wako) in serum-free medium. The cellular RNA was extracted by an RNeasy Mini Kit (Qiagen). The cDNA was synthesized with an RT^2^ First Strand cDNA Kit (Qiagen), according to the manufacturer’s protocol [10]. A TLR signaling pathway PCR array was purchased from Qiagen. A real-time PCR array based on the SYBR Green method was performed on a Quant Studio 3 real-time PCR system (Applied Biosystems, Foster, CA, USA). The cycling program was as follows: 95 °C for 10 min for 1 cycle, then 40 cycles of 95 °C for 15 s and 60 °C for 1 min. The housekeeping genes beta-2-microglobulin (B2M), hypoxanthine phosphoribosyltransferase 1 (HPRT1), ribosomal protein L13a (RPL13A), glyceraldehyde-3-phosphate dehydrogenase (GAPDH) and actin beta (ACTB) served as the internal controls. The data were analyzed by the Excel-based RT^2^ Profiler PCR Array Data Analysis software and website, which was provided by Qiagen (https://dataanalysis2.qiagen.com/pcr (accessed on 4 Juanury 2021)).

### 4.7. Reporter Assay for the Activation of the Interferon-β Promoter

IHH or Huh7 cells were transfected using Effectene transfection reagent, with plasmid DNAs (0.15 μg) encoding the firefly luciferase gene under the control of the IFN-β promoter (IFN-β-luc) and 25 μM si-MAP2K3 or 25 μM si-C, with or without poly(I:C) (Novus Biologicals, Littleton, CO, USA) (2.5 μg/mL). At 16 h post-transfection, the cells were lysed with reporter lysis buffer (Promega, WI, USA), and the luciferase activity was measured using a luminometer (Luminescencer-PSN AB-2200, ATTO, Tokyo, Japan).

### 4.8. Statistical Analysis

The data were expressed as the mean ± standard deviation. The comparisons were analyzed using Student’s *t* test. Significance was defined as a *p* < 0.05. The statistical analysis was conducted using the Excel Statistics program for Windows 2010 (SSRI, Tokyo, Japan) and DA stats software version PAF01644 (NIFTY Corp., Tokyo, Japan).

## 5. Conclusions

HAV occasionally causes acute liver failure, and HAV infection is one of the important insults in acute-on-chronic liver failure (ACLF). The severity of the ACLF caused by acute hepatitis A may be associated with severe underlying chronic liver diseases, such as advanced liver fibrosis and cirrhosis [4,38]. Some countries, such as Japan, have no universal vaccination program for HAV infection, and drug development for hepatitis A could be useful for controlling HAV infection and preventing severe hepatitis A. In conclusion, MAP2K3 could be one of the modulation factors for HAV PAMP triggers that produce interferon-β. We also found that MAP2K3 was a target candidate for drugs to inhibit HAV infection. The inhibition of MAP2K3 may prevent patients infected with HAV from developing severe hepatitis A.

## Figures and Tables

**Figure 1 ijms-22-07420-f001:**
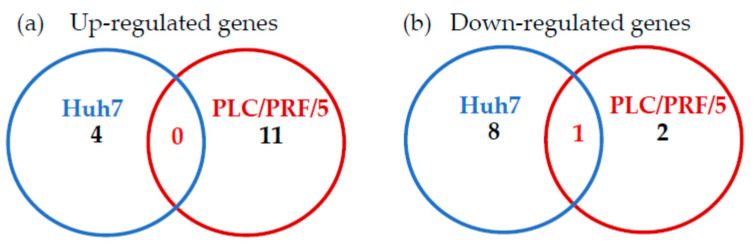
Differential expression of Toll-like receptor signaling pathway-related genes in Huh7 and PLC/PRF/5 cells treated with zinc chloride: (**a**) significantly upregulated genes compared to control; (**b**) significantly downregulated genes compared to control.

**Figure 2 ijms-22-07420-f002:**
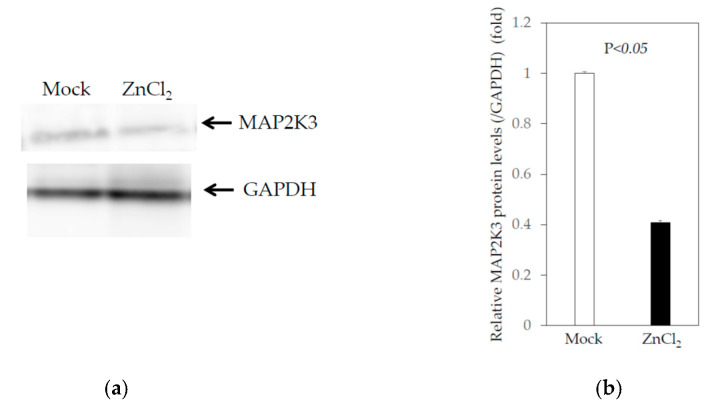
Knockdown of mitogen-activated protein kinase kinase 3 (MAP2K3) could lead to a decrease in HAV subgenomic and genomic RNA replication: (**a**) effect of 5 μM zinc chloride (ZnCl_2_) for 24 h on MAP2K3 and GAPDH expression at the protein level. A Western blot test demonstrated the MAP2K3 and GAPDH expression, using a specific antibody for MAP2K3 and for GAPDH, respectively; (**b**) the MAP2K3/GAPDH ratios from three independent experiments were measured using Adobe Photoshop CS5 (Adobe Inc., San Jose, CA, USA); (**c**) effect of the knockdown of MAP2K3 by specific siRNA compared with the control siRNA (si-C) on MAP2K3 and β-tubulin expression. A Western blot test demonstrated the MAP2K3 and β-tubulin expression using a specific antibody for MAP2K3 and for β-tubulin, respectively. (**d**) The MAP2K3/β-tubulin ratios from three independent experiments were measured; (**e**) the effect of the knockdown of MAP2K3 on the HAV genotype IB subgenomic replicon replication in HuhT7; (**f**) the effect of the knockdown of MAP2K3 on HAV genotype IIIA HA11-1299 replication in Huh7. The data are expressed as the means ± standard deviations of triplicate determinations from one experiment which was representative of three independent experiments. The original images for the blots (**a**,**c**) are shown in the Appendix A.

**Figure 3 ijms-22-07420-f003:**
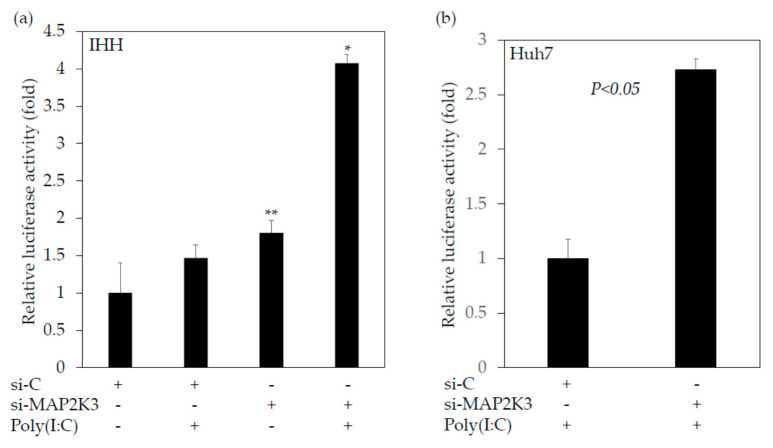
Knockdown of mitogen-activated protein kinase kinase 3 (MAP2K3) could enhance the interferon-β promoter activities stimulated by polyinosinic–polycytidylic acid (poly(I:C)). (**a**) IHH cells * *p* < 0.05 vs. other groups. ** *p* < 0.05 vs. the control siRNA (si-C) without poly(I:C) group; (**b**) Huh7 cells.

**Figure 4 ijms-22-07420-f004:**
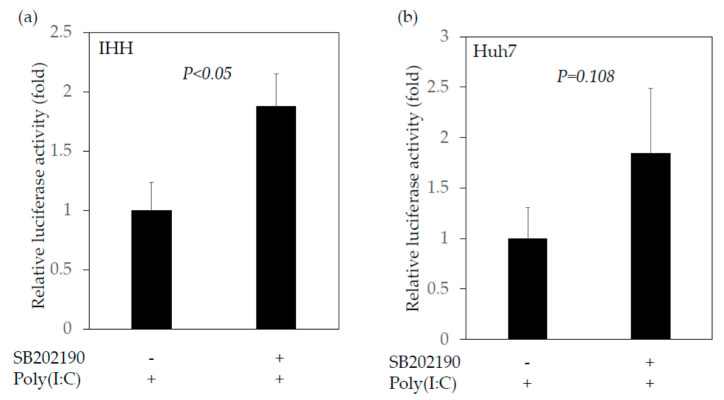
SB202190, a selective inhibitor of the MAP2K3–p38 MAPK signaling pathway, enhanced the interferon-β promoter activity stimulated by polyinosinic–polycytidylic acid (poly(I:C)). (**a**) IHH cells; (**b**) Huh7 cells. SB202190 was used at a concentration of 10 nM for 16 h.

**Figure 5 ijms-22-07420-f005:**
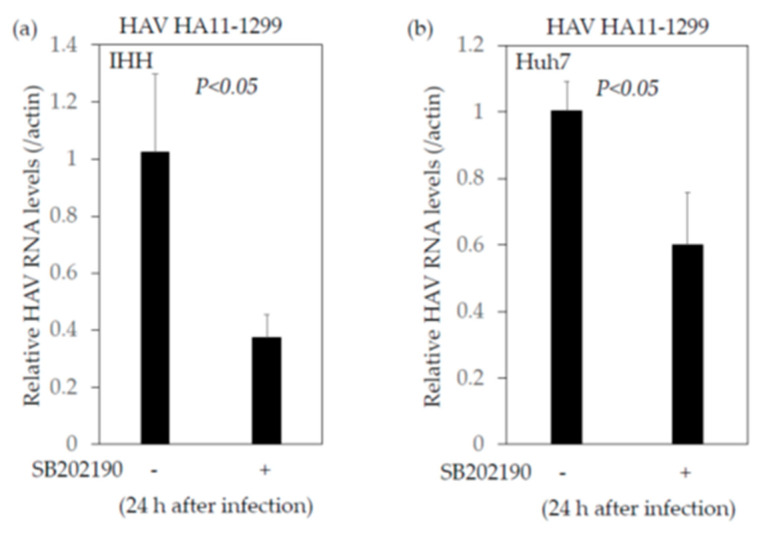
SB202190 inhibits hepatitis A virus (HAV) genotype IIIA HA 11-1299 replication. (**a**) IHH cells; (**b**) Huh7 cells. SB202190 was used at the concentration of 5 nM for 24 h.

**Table 1 ijms-22-07420-t001:** Toll-like receptor (TLR) signaling pathway-related genes significantly regulated by zinc chloride in Huh7 cells.

Gene Symbol	Roles of Genes	Fold Change ^1^	*p*-Values
(a) Upregulated Genes			
*NFKB2*	Signaling downstream of TLRs (NFκB signaling)	1.07	0.009920
*MAPK8IP3*	Signaling downstream of TLRs (JNK/p38 signaling); TLR interacting proteins and adaptors	1.11	0.007072
*IL8*	Pathogen-specific responses (bacterial response; fungal/parasitic response)	1.09	0.008839
*FOS*	Pathogen-specific responses (bacterial response); signaling downstream of TLRs (JNK/p38 signaling)	1.45	0.001211
(b) Downregulated genes			
*IL6*	Pathogen-specific responses (bacterial response; viral response); signaling downstream of TLRs (JAK/STAT signaling; cytokine signaling)	0.53	0.000928
*TLR5*	TLRs; TLR signaling (MYD88-dependent signaling)	0.57	0.044855
*IRF3*	Pathogen-specific responses (viral response); TLR signaling [TICAM1 (TRIF)-dependent (MYD88-independent) signaling]	0.77	0.001549
*RIPK2*	Pathogen-specific responses (bacterial response); TLR-interacting proteins and adaptors	0.80	0.015430
*TICAM1*	Pathogen-specific responses (bacterial response; viral response); TLR signaling [TICAM1 (TRIF)-dependent (MYD88-independent)]; TLR-interacting proteins and adaptors	0.82	0.003990
*PPARA*	Signaling downstream of TLRs (NFκB signaling); downstream effectors of TLR signaling	0.84	0.019047
*TICAM2*	TLR signaling [TICAM1 (TRIF)-dependent (MYD88-independent)]; TLR-interacting proteins and adaptors	0.89	0.001470
*HRAS*	Pathogen-specific responses (bacterial response; fungal/parasitic response); TLR-interacting proteins and adaptors	0.89	0.021661
*MAP2K3*	Signaling downstream of TLRs (JNK/p38 signaling)	0.90	0.034805

*NFKB2*, nuclear factor kappa B subunit 2; *MAPK8IP3*, mitogen-activated protein kinase 8 interacting protein 3; *IL8*, interleukin 8; *FOS*, Fos proto-oncogene, AP-1 transcription factor subunit; *IL6*, interleukin 6; *TLR5*, Toll-like receptor 5; *IRF3*, interferon regulatory factor 3; *RIPK2*, receptor interacting serine/threonine kinase 2; *TICAM1*, Toll-like receptor adaptor molecule 1; *PPARA*, peroxisome proliferator activated receptor alpha; *TICAM2*, Toll-like receptor adaptor molecule 2; *HRAS*, HRas proto-oncogene, GTPase; *MAP2K3*, mitogen-activated protein kinase kinase 3; ^1^ Fold change (cellular RNA extracted from Huh7 cells treated with zinc chloride (*n* = 3)/untreated control (*n* = 3)).

**Table 2 ijms-22-07420-t002:** Toll-like receptor (TLR) signaling pathway-related genes significantly regulated by zinc chloride in PLC/PRF/5 cells.

Gene Symbol	Roles of Genes	Fold Change ^1^	*p*-Values
(a) Upregulated Genes			
*EIF2AK2*	Pathogen-specific responses (viral response); downstream effectors of TLR Signaling	1.07	0.030044
*PPARA*	Signaling downstream of TLRs (NFκB signaling); downstream effectors of TLR signaling	1.08	0.028574
*HMGB1*	Pathogen-specific responses (bacterial response); TLR-interacting proteins and adaptors	1.12	0.028282
*SARM1*	TLR signaling (negative regulation of TLR signaling); TLR-interacting proteins and adaptors	1.12	0.007063
*MAP2K4*	Signaling downstream of TLRs (JNK/p38 signaling)	1.13	0.019787
*REL*	Signaling downstream of TLRs (NFκB signaling)	1.15	0.019521
*HRAS*	Pathogen-specific responses (bacterial response; fungal/parasitic response); TLR-interacting proteins and adaptors	1.15	0.007300
*MAPK8*	Signaling downstream of TLRs (JNK/p38 signaling)	1.19	0.000817
*IRAK1*	Pathogen-specific responses (bacterial response); TLR signaling (MYD88-dependent signaling); signaling downstream of TLRs (NFκB signaling; cytokine signaling); downstream effectors of TLR signaling	1.29	0.004801
*TIRAP*	Pathogen-specific responses (fungal/parasitic response); TLR signaling (MYD88-dependent signaling); TLR-interacting proteins and adaptors	1.32	0.035440
*IL1A*	Signaling downstream of TLRs (cytokine signaling)	1.45	0.024464
(b) Downregulated genes			
*SIGIRR*	TLRs; TLR signaling (negative regulation of TLR signaling); signaling downstream of TLRs (cytokine signaling)	0.62	0.034271
*MAP2K3*	Signaling downstream of TLRs (JNK/p38 signaling)	0.86	0.000931
*MYD88*	TLR signaling (MYD88-dependent); TLR-interacting proteins and adaptors	0.93	0.028868

*EIF2AK2*, eukaryotic translation initiation factor 2 alpha kinase 2; *PPARA*, peroxisome proliferator activated receptor alpha; *HMGB1*, high mobility group box 1; *SARM1*, sterile alpha and TIR motif containing 1; *MAP2K4*, mitogen-activated protein kinase kinase 4; *REL*, REL proto-oncogene, NF-kB subunit; *HRAS*, HRas proto-oncogene, GTPase; *MAPK8*, mitogen-activated protein kinase 8; *IRAK1*, interleukin 1 receptor associated kinase 1; *TIRAP*, TIR domain containing adaptor protein; *IL1A*, interleukin 1 alpha; *SIGIRR*, single Ig and TIR domain containing; *MAP2K3*, mitogen-activated protein kinase kinase 3; *MYD88*, MYD88 innate immune signal transduction adaptor; ^1^ Fold change (cellular RNA extracted from PLC/PRF/5 cells treated with zinc chloride (*n* = 3)/untreated control (*n* = 3)).

## Data Availability

The data underlying this article are available in this article.

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
