# Peer review of "Knockdown of Mitogen-Activated Protein Kinase Kinase 3 Negatively Regulates Hepatitis A Virus Replication"

_ijms, 2021, doi:10.3390/ijms22147420_

Round 1

Reviewer 1 Report

In this study titled ‘Knockdown or Mitogen-Activated Protein Kinase Kinase 3 Negatively Regulates Hepatitis A Virus Replication’ the authors examined the TLR-signaling associated genes differentially expressed in human hepatoma cells when treated with zinc chloride. They selected MAP2K3 as a target for further analyses. Gene perturbation studies using MAP2K3 siRNA showed reduction in MAP2K3 protein levels and the HAV replication was reduced ~50% in MAP2K3 depleted cells.

Overall, the quality of the data presented is preliminary and the findings need a significant amount of supporting evidence before a concrete conclusion regarding the role of MAP2K3 in HAV life cycle and its potential as a therapeutic target can be established. I think the study in the current form is very preliminary to be published in its present form. I recommend the authors to do consider the points outlined below and design further experiments to improve the quality of the manuscript before resubmission.

Major points

  1. The rationale behind why the authors picked MAP2K3 as their candidate for further analysis is confusing. According to the fold change calculation formula (in legend for table 1), there was only 10-14% reduction in levels of MAP2K3 mRNA in their zinc chloride treatment experiment in both cell lines. The authors failed to explain why this candidate was picked when this showed the weakest phenotype among all their hits.
  2. Given the very low reduction in mRNA levels of MAP2K3, the authors should check the MAP2K3 protein levels in zinc chloride treated cells to validate their mRNA change findings. If they could not detect significant drop in protein level in cells, all further characterization of MAP2K3 is logically invalid.
  3. The quality of the blot in fig 2A is low with uneven staining of the MAP2K3 bands, and there is no quantitation or statistical data for a minimum of three independent experiments presented. The authors have to show a minimum of two independent siRNAs for MAP2K3, or provide protein reconstitution experiment data or use chemical inhibitors to rule out that the observed phenotype is not an off-target effect of the siRNA used.
  4. As cancer cells and cell lines are known to be more sensitive to silencing of MAP2K3, the authors have to perform gene perturbation studies in primary human hepatoma cells to rule out that the phenotype is not a cell line artefact.
  5. To argue any therapeutic potential of MAP2K3 as a druggable target, the authors have to show significant effect when chemical inhibitors against MAP2K3 is used in a primary cell context or in an animal model.

Author Response

Response to Reviewers’ Comments: Thank you very much for your invaluable comments.

Response to Comments of Reviewer #1: Thank you very much for your encouraging comments. We revised our manuscript accordingly.

In Abstract section,

“… inhibition of MAP2K3. In conclusion, MAP2K3 could be a modulation factor for HAV pathogen associated molecular pattern (PAMP) triggers production of interferon-β. MAP2K3 seems to play a role in antiviral immunity against HAV infection. MAP2K3 is one of the promising targets for drugs that fight against HAV infection. The inhibition of MAP2K3 may prevent patients infected with HAV from developing severe hepatitis A.”

In Conclusions section, page 11, lines 346-347,

“…severe hepatitis A. In conclusion, MAP2K3 could be one of the modulation factors for HAV PAMP triggers production of interferon-β. We also found that MAP2K3 is one candidate for the…”

Response to Major point 1 of Reviewer #1: “The rationale behind why the authors picked MAP2K3 as their candidate for further analysis is confusing. According to the fold change calculation formula (in legend for table 1), there was only 10-14% reduction in levels of MAP2K3 mRNA in their zinc chloride treatment experiment in both cell lines. The authors failed to explain why this candidate was picked when this showed the weakest phenotype among all their hits.”

Thank you very much for your invaluable comments. We revised our manuscript accordingly.

In Result section, page 4, lines 118-124,

2.3.1. Knockdown of MAP2K3 by siRNA did not reduce cell viabilities of Huh7 cells

The induction factors of the genes studied following treatment of human hepatoma cell lines with zinc chloride indicated ranged from 1.07 to 1.45 which seems relatively low, and likewise for the downregulation factors, all less than 2 even if statistically significant (Tables 1 and 2). Therefore, we also examined whether MAP2K3 protein expression was downregulated in Huh7 cells treated with 5 μM of zinc chloride for 24 h (Figure 2a, 2b). Thus, we also confirmed that zinc chloride downregulated MAP2K3 protein expression at protein level.

Next, we transfected…

Response to Major point 2 of Reviewer #1: “Given the very low reduction in mRNA levels of MAP2K3, the authors should check the MAP2K3 protein levels in zinc chloride treated cells to validate their mRNA change findings. If they could not detect significant drop in protein level in cells, all further characterization of MAP2K3 is logically invalid.”

Thank you very much for your invaluable comments. We revised our manuscript accordingly.

In Result section, page 4, lines 118-124,

2.3.1. Knockdown of MAP2K3 by siRNA did not reduce cell viabilities of Huh7 cells

The induction factors of the genes studied following treatment of human hepatoma cell lines with zinc chloride indicated ranged from 1.07 to 1.45 which seems relatively low, and likewise for the downregulation factors, all less than 2 even if statistically significant (Tables 1 and 2). Therefore, we also examined whether MAP2K3 protein expression was downregulated in Huh7 cells treated with 5 μM of zinc chloride for 24 h (Figure 2a, 2b). Thus, we also confirmed that zinc chloride downregulated MAP2K3 protein expression at protein level.

Next, we transfected…

We also added new Figure 2 (a) and 2 (b).

Response to Major point 3 of Reviewer #1: “The quality of the blot in fig 2A is low with uneven staining of the MAP2K3 bands, and there is no quantitation or statistical data for a minimum of three independent experiments presented. The authors have to show a minimum of two independent siRNAs for MAP2K3, or provide protein reconstitution experiment data or use chemical inhibitors to rule out that the observed phenotype is not an off-target effect of the siRNA used.”

Thank you very much for your invaluable comments. We revised our manuscript accordingly.

We made a change to a new blot in Figure 2 (c) of the revised manuscript, and demonstrated protein reconstitution experiment data in Figure 2 (d) of the revised manuscript. We used SB202190, a selective inhibitor of MAP2K3-p38 MAPK signaling pathway in several experiments and showed these data.

Response to Major point 4 of Reviewer #1: “As cancer cells and cell lines are known to be more sensitive to silencing of MAP2K3, the authors have to perform gene perturbation studies in primary human hepatoma cells to rule out that the phenotype is not a cell line artefact.”

Thank you very much for your invaluable comments. We used immortalized human hepatocytes as well as Huh7 cells to examine the study of “activation of interferon-β promoter activities stimulated by poly(I:C)” to rule out that the phenotype is not a cell line artefact. Werevised our manuscript accordingly.

Response to Major point 5 of Reviewer #1: “To argue any therapeutic potential of MAP2K3 as a druggable target, the authors have to show significant effect when chemical inhibitors against MAP2K3 is used in a primary cell context or in an animal model.”

Thank you very much for your invaluable comments. We examined whether SB202190, a potent inhibitor of MAP2K3-p38 MAPK signaling pathway enhances interferon-β promoter activities stimulated by poly(I:C) and whether SB202190 inhibits HAV replication or not. We used IHH cells as non-cancerous hepatocytes as well as Huh7 cells. We revised our manuscript accordingly and added new Figures 4 and 5.

Reviewer 2 Report

Kanda and colleagues examined the effects of zinc chloride on TLR signaling pathways in human hepatoma cell lines. They observed that zinc chloride inhibits MAP2K3 expression leading to a deacrease of HAV replication. This is an interesting observation, the experiments are well conducted and the paper clearly written.

My comments are the following :

  • The induction factors of the genes studied following treatment of human hepatoma cell lines with zinc chloride indicated in table 1 and 2 ranged from 1.07 to 1.45 which seems very low even if statistically significant. Likewise for the downregulation factors, all less than 2. It would be interesting to study these modulation factors in a context of stimulation of the cell by a TLR agonist such as Poly I: C.
  • The role of MAP2K3 in inhibiting HAV replication would desserve further discussion. Moreover, the conclusion does not describe the possible consequences of this interesting observation. 

Author Response

Response to Comments of Reviewer #2: Thank you very much for your encouraging comments.

Response to Major comment 1 of Reviewer #2: “The induction factors of the genes studied following treatment of human hepatoma cell lines with zinc chloride indicated in table 1 and 2 ranged from 1.07 to 1.45 which seems very low even if statistically significant. Likewise for the downregulation factors, all less than 2. It would be interesting to study these modulation factors in a context of stimulation of the cell by a TLR agonist such as Poly I: C.”

Thank you very much for your invaluable comments. We extensively revised our manuscript accordingly, revised new Figure 2, and added Figures 3, 4, and 5.

Response to Major comment 2 of Reviewer #2: “The role of MAP2K3 in inhibiting HAV replication would desserve further discussion. Moreover, the conclusion does not describe the possible consequences of this interesting observation.”

Thank you very much for your invaluable comments. We extensively revised our manuscript accordingly, revised new Figure 2, and added Figures 3, 4, and 5.

In Conclusions section, page 11, lines 346-347,

“…severe hepatitis A. In conclusion, MAP2K3 could be one of the modulation factors for HAV PAMP triggers production of interferon-β. We also found that MAP2K3 is one candidate for the…”

Reviewer 3 Report

Kanda et al show that out of 84 TLR signaling pathway-related genes examined, MAP2K3 was the only significantly downregulated gene in both Huh7 and PLC/PRF/5 cell lines. Therefore, the authors focused on the association between MAP2K3 and HAV replication.

I have a few concerns with this manuscript:

  1. Please explain the rationale why just Huh7 and PLC/PRF/5 cell lines were selected for this study. Not surprisingly the gene expression in both cell lines differed upon zinc chloride treatment in most genes involved in TLR signaling. Primary human hepatocytes should be used to avoid possible artificial effects of cell lines.
  2. It was shown previously in a study by Chen et al (PLoS Pathog 9(4): e1003248 (2013)) performed in Huh7 cell line, that activation of MAPK and FOS results in upregulation of miR-21, which downregulates MyD88 and IRAK and leads to inhibition of type I IFN production. Please interpret your results with MAP2K3 in the context of these findings. Several other papers show that inhibition of MAPK upregulate type I IFN production.
  3. Effect of 5 microM zinc chloride on HAV replication was investigated in this and previous studies of the authors of this manuscript. How relevant is this concentration for preclinical or clinical studies of inhibition of HAV infection considering that zinc chloride will be used as adjuvant of some other therapy?

Minor points

  1. Figure 2: How many times was the experiment repeated? Is the data distribution normal and is the use of parametric Student test for statistic evaluation substantiated?  Are the differences still significant when non-parametric test will be used?
  2. Figure 2a. Please provide a better-quality Western blot of MAP2K3.

Author Response

Response to Comments of Reviewer #3: Thank you very much for your invaluable comments.

Response to Concern 1 of Reviewer #3: “Please explain the rationale why just Huh7 and PLC/PRF/5 cell lines were selected for this study. Not surprisingly the gene expression in both cell lines differed upon zinc chloride treatment in most genes involved in TLR signaling. Primary human hepatocytes should be used to avoid possible artificial effects of cell lines.”

Thank you very much for your invaluable comments. As Huh7 and PLC/PRF/5 cells support HAV replication [9, 33, 38], we selected these cell lines in this study. We did not use primary human hepatocytes, but, to avoid possible artificial effects of cell lines, we used immortalized human hepatocytes in several experiments. We extensively revised our manuscript accordingly.

Response to Concern 2 of Reviewer #3: “It was shown previously in a study by Chen et al (PLoS Pathog 9(4): e1003248 (2013)) performed in Huh7 cell line, that activation of MAPK and FOS results in upregulation of miR-21, which downregulates MyD88 and IRAK and leads to inhibition of type I IFN production. Please interpret your results with MAP2K3 in the context of these findings. Several other papers show that inhibition of MAPK upregulate type I IFN production.”

Thank you very much for your invaluable comments. We extensively revised our manuscript accordingly.

In Discussion section, page 9, lines 240-244,

…demonstrated that siRNA against MAP2K3 inhibits HAV replication (Figure 2). Chen et al. also reported that activation of MAPK and Fos proto-oncogene (FOS) results in upregulation of miR-21, which downregulates MyD88 and interleukin 1 receptor associ-ated kinase 1 (IRAK1), and leads to inhibition of type I IFN production in Huh7 cells [35]. Our results support these findings [35]. Therefore, MAP2K3 seems to play a role in the antiviral immunity towards HAV infection….

Response to Concern 3 of Reviewer #3: “Effect of 5 microM zinc chloride on HAV replication was investigated in this and previous studies of the authors of this manuscript. How relevant is this concentration for preclinical or clinical studies of inhibition of HAV infection considering that zinc chloride will be used as adjuvant of some other therapy?”

Thank you very much for your invaluable comments. We extensively revised our manuscript accordingly.

In Discussion section, page 9, lines 246-253,

In Japan, zinc chloride is not under clinical use for acute HAV infection. Normal serum zinc concentration is from 122 to 199 μM [36]. In Japanese daily clinical practice, patients with zinc deficiency or Wilson disease take zinc acetate including 50-100 mg daily zinc or 150 mg daily zinc. Zinc deficiency is often observed in patients with liver diseases [36]. Zinc plays a role in the activation and structural maintenance of as many as 300 proteins and enzymes that contribute to various biological processes [36]. Acute HAV infection occasionally leads to acute liver failure with coagulopathy and hyperammonemia [37]. Zinc supplementation is also useful for improve hyperammonemia [36].

Response to Minor comment 4 of Reviewer #3: “Figure 2: How many times was the experiment repeated? Is the data distribution normal and is the use of parametric Student test for statistic evaluation substantiated? Are the differences still significant when non-parametric test will be used?”

Thank you very much for your invaluable comments. Data are expressed as means ± standard deviations of triplicate determinations from 1 experiment representative of three independent experiments. The differences still significant when non-parametric test was used. We also demonstrated the protein data in Figure 2 (a) and 2 (b). We extensively revised our manuscript.

 Response to Minor comment 5 of Reviewer #3: “Figure 2a. Please provide a better-quality Western blot of MAP2K3.”

Thank you very much for your invaluable comments. We revised new Figure 2 (b) and 2 (c) of the revised manuscript, and we also extensively revised our manuscript accordingly.

Round 2

Reviewer 1 Report

The authors have not sufficiently addressed several concerns raised by this reviewer. So I would recommend the authors to address the concerns raised below, and resubmit the manuscript with requested data and corrections.

  1. The first question raised was how the MAP2K3 candidate was picked when the fold change was 0.9 which means a reduction of 10% at RNA level. The authors added a paragraph of text (lines 118-124) to clarify their selection process, but in reality did not explain anything new. As far as this reviewer understood, compared to control cells, the candidate (MAP2K3) had a reduction of 10 and 14 percentage in Huh7 and PLC/PRF/5 cells respectively following treatment with Zinc Chloride. Do the authors agree with this statement?
  2. The quality of Western Blot in fig 2A requires improvement and a more representative data should be provided. The 50-60% reduction the authors quantitated (fig 2b) is not evident in fig 2a. As data in Fig 2B and Fig 2D are normalized to control, that information has to be added to the Y-axis. The statement in figure legend 2 starting with "Data are expressed as means"(lines 140-141) is very confusing and need to be re-worded to clearly state the number of biological replicates and the statistical analysis used.
  3. To check whether MAP2K3 is a negative regulator of IFN induction, they authors used a reporter assay (Fig. 3). They should directly measure the IFN-B transcript levels and IFN-B in culture supernatant by ELISA to further validate this mechanistic link.
  4. As Huh7 cells are not known to produce significant amount of type-I IFNs, the  inhibition in HAV replication  seen in these cells when treated with p38-MAPK inhibitor SB202190 is most likely due to some other mechanism. The authors have to discuss this possibility in their results and discusion section. This point is further reinforced by the observation that the authors saw no significant increase in reporter activity in Fig.4. in Huh7 cells.
  5. As enhancement of IFN production is proposed as the main mode of action of SB202190 against HAV, they should show the level of secreted IFN-B by ELISA to further confirm the point.

Author Response

(Response to Reviewer 1)

Response to Comments of Reviewer #1: Thank you very much for your invaluable comments.

Response to Comment 1 of Reviewer #1: “The first question raised was how the MAP2K3 candidate was picked when the fold change was 0.9 which means a reduction of 10% at RNA level. The authors added a paragraph of text (lines 118-124) to clarify their selection process, but in reality did not explain anything new. As far as this reviewer understood, compared to control cells, the candidate (MAP2K3) had a reduction of 10 and 14 percentage in Huh7 and PLC/PRF/5 cells respectively following treatment with Zinc Chloride. Do the authors agree with this statement?”

Thank you very much for your comments. We agree with you. According to your suggestions, we extensively revised our manuscript.

Response to Comment 2 of Reviewer #1: “The quality of Western Blot in fig 2A requires improvement and a more representative data should be provided. The 50-60% reduction the authors quantitated (fig 2b) is not evident in fig 2a. As data in Fig 2B and Fig 2D are normalized to control, that information has to be added to the Y-axis. The statement in figure legend 2 starting with "Data are expressed as means"(lines 140-141) is very confusing and need to be re-worded to clearly state the number of biological replicates and the statistical analysis used.”

Thank you very much for your comments. According to the comments of Review Report (Round 1), we have revised as “Data are expressed as means ± standard deviations of triplicate determinations from 1 experiment representative of three independent experiments.” and we added “Figure 2”. So, we cannot make change of these parts as Reviewer 3 gave us the comment as “Paper improved significantly.” We are sorry for Reviewer 1. But we extensively revised our figure 2b and 2d, according to your suggestions.

Response to Comment 3 of Reviewer #1: “To check whether MAP2K3 is a negative regulator of IFN induction, they authors used a reporter assay (Fig. 3). They should directly measure the IFN-B transcript levels and IFN-B in culture supernatant by ELISA to further validate this mechanistic link.”

Thank you very much for your comments. We previously demonstrated IFN Bioassay using IHH cells [Ref.20]. Because our experiment were complex, for example, luciferase gene, poly (I:C), siRNAs, we cannot perform IFN Bioassay, which reflects more precise IFN in culture supernatant. We think IFN ELISA cannot measure the bioactive of IFN. According to your suggestions, we extensively revised our manuscript.

Response to Comment 4 of Reviewer #1: “As Huh7 cells are not known to produce significant amount of type-I IFNs, the inhibition in HAV replication seen in these cells when treated with p38-MAPK inhibitor SB202190 is most likely due to some other mechanism. The authors have to discuss this possibility in their results and discusion section. This point is further reinforced by the observation that the authors saw no significant increase in reporter activity in Fig.4. in Huh7 cells.”

Thank you very much for your comments. We agree with you. According to your suggestions, we revised our manuscript.

Response to Comment 5 of Reviewer #1: “As enhancement of IFN production is proposed as the main mode of action of SB202190 against HAV, they should show the level of secreted IFN-B by ELISA to further confirm the point.”

Thank you for your comments. We agree with you, to some extents. Please see our responses to your comments 3 and 4. We extensively revised our manuscript.

Reviewer 3 Report

Paper improved significantly.

Author Response

Response to Comments of Reviewer #3: Thank you very much for your encouraging comments.
